# Uneven economic burden of non-communicable diseases among Indian households: A comparative analysis

**Sasmita Behera** **\*, Jalandhar Pradhan**

Department of Humanities and Social Sciences, National Institute of Technology, Rourkela, India

\* sasmitabehera483@gmail.com

## Abstract

### Background

Non-communicable diseases (NCDs) are the leading global cause of death and disproportionately concentrate among those living in low-income and middle-income countries. However, its economic impact on households remains less well known in the Indian context. This study aims to assess the economic impact of NCDs in terms of out-of-pocket expenditure (OOPE) and its catastrophic impact on NCDs affected households in India.

### Materials and methods

Data were collected from the 75th round of the National Sample Survey Office, Government of India, conducted in the year 2017–18. This is the latest round of data available on health, which constitutes a sample of 113,823 households. The collection of data is based on a stratified multi-stage sampling method. Generalised Linear Regression model was employed to identify the socio-economic covariates associated with the catastrophic health expenditure (CHE) on hospitalisation.

### Results

The result shows a higher burden of OOPE on NCDs affected households. The mean expenditure by NCDs households in public hospitals is INR 13,170 which is more than twice as compared to the non-NCDs households INR 6,245. Particularly, the proportion of total medical expenditure incurred on medicines (0.39) and diagnostics (0.15) is troublesome for households with NCDs, treated in public hospitals. Moreover, results from the generalised linear regression model confirm the significant relationship between CHE with residence, caste, religion, household size, and economic status of households. The intensity of CHE is more for the households who are poor, drinking unsafe water, using firewood as cooking fuel, and household size of 1–5 members.

### Conclusion

Therefore, an urgent need for a prevention strategy should be made by the government to protect households from the economic burden of NCDs. Specifically, to reduce the burden

**Data Availability Statement:** Data are available in the website of Ministry of Statistics and Programme Implementation (MOSPI), government of India. It can be retrieved from http://mospi.nic.in/unit-level-data-report-nss-75th-round-july-2017-

june-2018-schedule-250social-consumption-health.

**Funding:** The authors received no specific funding for this work.

**Competing interests:** The authors have declared that no competing interests exist.

of CHE associated with NCDs, a customised disease-specific health insurance package should be introduced by the government of India in both public and private facilities.

## Introduction

The escalating burden of non-communicable diseases (NCDs) is presently being experienced by all the countries across the globe [1, 2]. However, the disproportionate concentration of this burden is well documented in the case of low-income and middle-income countries where NCDs kill 15 million people every year, out of which 85% are premature deaths [3]. NCDs also account for 58% of Disability Adjusted Life Years (DALYs) [4]. The adverse impact of NCDs is a growing concern for developing countries where public spending on NCDs is relatively scant, and people have limited resources to accommodate their healthcare needs [5, 6]. In the absence of an adequate financial mechanism, out-of-pocket expenditure (OOPE) for the treatment and care of NCDs often traps the households in the cycle of catastrophic expenditure that forces the households to various financial shocks such as borrowing money or (and) selling assets [7, 8]. NCDs curtail household income and a family's ability to spend on its basic necessities like food, clothing, and education expenditure [9–11]. The economic cost of NCDs has also a significant macroeconomic effect on the Indian economy. NCDs reduce the productivity of the workforce, resulting in the reduction of overall economic output [12, 13]. It is estimated that every 10% increase in NCDs mortality results in a 0.5% reduction in annual economic growth. [14].

The increasing burden of OOPE confronted two important aspects of Universal Health Coverage (UHC): first, every population, irrespective of rich and poor should get needed health care services, not only those who can afford it (equity prospective) [15, 16]; second, the cost of health care should not put people at the risk of any financial hardship (financial risk protection) [17]. The burden of NCDs further puts a grave threat to Sustainable Development Goals (SDGs-3.4) i.e. to reduce one-third premature mortality from NCDs. The health financing system should be focused on the aspect of equity as well as provide financial protection to the poor. Budgetary allocation for healthcare spending is also an important aspect to look forward to lessen the devastating burden of NCDs on households.

India is experiencing the rising burden of NCDs, with limited access to health care and social security [18, 19]. The World Health Organisation (WHO) estimates that NCDs account for 63% of all deaths, out of which 27% of the deaths are from cardiovascular disease, 9% from cancer, 3% from diabetes, and 11% from chronic respiratory disease in India [20]. Apart from mortality, NCDs burden can also be well captured in terms of DALYs where it captures not only mortality but also years of productive life lost due to premature mortality and years with disability. As per the study by the Indian Council of Medical Research, in India over the years 1990 to 2016, the proportion of total DALY attributable to NCDs increased from 30.5% to 55.4% [21].

National Health Account estimates that OOPE constitutes 58.7% of total health expenditure in India [22]. However, India is spending only 1.2% of its Gross Domestic Product (GDP) on health expenditure for the year 2016–17 [22]. Due to the lack of a better financial mechanism in India, excessive dependence of households on OOPE occurs for treatment of NCDs (including medication, diagnostic test, and drug therapy) that forces 8.50% of people to below the poverty line in the year 2014, which again increase to 12.43% in 2017–18 [23]. An epidemiological assessment of OOPE associated with NCDs households and their distribution across

various socio-economic groups in India would provide policymakers with additional information about the households' spending patterns, which could help them in formulating effective and multisectoral interventions to reduce household financial burden.

Based on the available literature, it is clear that although there is a plethora of literature available that gives us the evidence of poverty impact of OOPE in the context of health care financing [24–28], the impact of NCDs on OOPE experienced by households is poorly researched, particularly in the Indian context [29, 30]. Further, the existing studies are limited to some particular types of NCDs like diabetes, hypertension, stroke, and cancer. The present study attempts to fill these gaps by investigating the NCDs attributable OOPE and its catastrophic impact on Indian households by focusing on a group of NCDs. Further, it will give a comparative analysis of OOPE experienced by both NCDs and non-NCDs prevalence households. The study also provides an idea about the type of healthcare facility (public/private) people are choosing for treatment of NCDs as well as associate OOPE from each type of healthcare facility in India.

## Materials and methods

### Data

The present study is based on household social consumption on health data, collected by National Sample Survey Office (NSSO), Government of India, in the year 2017–18 [31]. This is the latest round of data available on health, which constitutes a sample of 113,823 households. The collection of data is based on a stratified multi-stage sampling method. In this survey, an investigation was made to know the nature of ailment for those people who are hospitalised, the extent to which people are using public and private hospitals, and the amount of expenditure incurred for the treatment received from both government and private healthcare facilities. The recall period in the survey was 365 days for inpatient care and 15 days for outpatient care. However, the present study is based on inpatient care that takes into account hospitalisation cases only. Hospitalisation is defined as an overnight stay in the hospital any time in 365 days prior to the survey date. Expenditure on hospitalization has been recorded separately under three broad categories: medical, non-medical, and transport expenditure. The medical expenditure includes doctor's fee, purchase of medicine and drugs, clinical test (x-ray, ECG, scan), bed charges, and other medical expenses like physiotherapy, blood, oxygen, etc. Similarly, the non-medical expenses include expenditure on food, escort, lodging charges, etc. Adding together medical, non-medical, and transport expenditures provide total expenditure on hospitalisation.

### Study design

The study is based on household level analysis where the aim is to compare the economic burden of OOPE among households associated with the prevalence of NCDs and non-NCDs. A household can be considered as an NCDs household if at least one of its members reported having one or more NCDs within the last 365 days. The NSSO (75th round) data provides a list of 60 diseases under 15 broad categories of infection, cancer, psychiatric disorder, respiratory, gastrointestinal, blood disease, metabolic and nutritional, obstetric, cardiovascular, skin, injury, genito-urinary, eyes, ear, and musculoskeletal. For our analysis, we have grouped 60 diseases into two broad categories, based on the International classification of diseases (ICD-10), where the first category includes only NCDs and the rest of all infectious/ communicable/ nutritional diseases/ injuries are grouped into the second category. Accordingly, we have categorized households with NCDs (the first group) and households with non-NCDs (the second

group). Out of the 56,731 households with hospitalisation cases, the number of NCDs house-holds is 21,776 and non-NCDs households are 34,955.

## Variables

The outcome variables of the study are OOPE and catastrophic health expenditure (CHE) for hospitalization by NCDs and non-NCDs households. The OOPE is further classified as medical, non-medical, transport, and other expenses by types of services (public/private). We calculate each component of OOPE separately, for both public and private facilities. The selection of independent variables is based on the past literature that shows socio-economic and demographic variables like caste, religion, economic capability, age, etc. have a significant impact on the pattern of healthcare spending, especially in the Indian context [32, 33]. Therefore, we investigated the hypothesis of whether these variables have an impact on OOPE. The independent variables included in the analysis are, the number of elderly (60+) in the household (no elderly, only one elderly, two or more than two elderly), the place of residence (rural and urban), religion (Hindu, Muslim, and others), caste (SC, ST, and Others), household size (1–5, 6–7, and 8+members), cooking fuel (firewood, LPG & other gases, and others), drinking water (safe and unsafe), type of latrine [(flush/pit), (open space /others)], wealth index (Poorest, Poorer, Middle, Richer and Richest), and region (North, East, Northeast, Central, West, and South).

## Method

The comparison between NCDs households and non-NCDs households has been made on the basis of the following grounds: (a) socio-economic factors determining the prevalence of diseases experienced by NCDs and non-NCDs households, (b) mean OOPE by type of health care facility (public/private), (c) proportion of households experiencing CHE, and (d) estimation of Generalised Linear Regression Model (GLM) to calculate the effect of various independent variables on the level of CHE. To measure OOPE we have followed the approach recommended by World Bank [34]. Here, OOPE is the share of health spending by the patient themselves at the time of receiving care. It can be of different forms, such as user fees which are directly paid to the healthcare provider at public facilities; co-payments which are paid by the insured person to the insurer; and payments made by individuals to private health care providers, not covered by any form of health insurance [35, 36]. Similarly, when the level of health expenditure exceeds some fraction of the household's total income/expenditure is known as CHE [35–37]. In the present study, households spending more than 10 percent of their total consumption expenditure on health are considered to be catastrophic. This is because the 10% threshold is the most widely used threshold level [35, 38–40].

## Results

### Socio-economic profile of sample households

Table 1 provides a descriptive analysis of households that shows out of 56,731 hospitalisation cases, 21,776 households were hospitalised due to NCDs, and 34,955 households hospitalised for disability, communicable, and infectious diseases. Households having more than two elderly reported more hospitalisation cases due to NCDs and fewer cases reported for non-NCDs. About 62% of households hospitalised due to NCDs are from rural areas and 37% are from urban areas. Similarly, 67% of non-NCDs households are from rural areas and 33% are from urban areas. Nearly 80% of households, both NCDs and non-NCDs, belong to the Hindu religion, 14% belong to Muslims, and the rest of the 6% belongs to other categories. The

**Table 1.  Socio-economic characteristics of households, NSSO survey, 2017–18.**

| Household's Characteristic | NCDs | NON-NCDs | TOTAL |
|---|---|---|---|
| Elderly members | | | |
| No elderly | 58.70 | 70.26 | 79.01 |
| One elderly | 25.54 | 19.62 | 17.70 |
| 2+ elderly | 15.76 | 10.12 | 3.29 |
| Residence | | | |
| Rural | 62.49 | 66.68 | 65.01 |
| Urban | 37.51 | 33.32 | 34.99 |
| Religion | | | |
| Hindu | 79.54 | 80.11 | 79.89 |
| Muslim | 14.07 | 14.01 | 14.03 |
| Others | 6.39 | 5.88 | 6.08 |
| Caste | | | |
| SC/ST | 23.40 | 26.73 | 25.41 |
| OBC | 42.76 | 43.86 | 43.42 |
| Others | 33.84 | 29.41 | 31.17 |
| Household size | | | |
| 1–5 | 66.43 | 70.97 | 69.16 |
| 6–7 | 22.51 | 20.21 | 21.12 |
| 8+ | 11.06 | 8.82 | 9.71 |
| Type of Cooking fuel | | | |
| Firewood | 34.38 | 36.01 | 35.36 |
| LPG/Other gas | 60.80 | 59.01 | 59.64 |
| Other | 4.82 | 4.98 | 5.00 |
| Drinking water | | | |
| Safe | 96.48 | 96.53 | 96.51 |
| Unsafe | 3.52 | 3.47 | 3.49 |
| Type of latrine | | | |
| Flush/pit | 82.02 | 78.88 | 80.13 |
| Open space/others | 17.98 | 21.12 | 19.87 |
| Wealth quintile | | | |
| Poorest | 14.89 | 14.86 | 14.87 |
| Poorer | 17.87 | 16.63 | 17.12 |
| Middle | 18.02 | 19.35 | 18.82 |
| Richer | 20.56 | 22.13 | 21.50 |
| Richest | 28.67 | 27.03 | 27.68 |
| Region | | | |
| North | 13.87 | 13.70 | 13.76 |
| Central | 19.72 | 19.79 | 19.76 |
| East | 20.35 | 20.29 | 20.31 |
| Northeast | 2.05 | 2.68 | 2.43 |
| West | 15.79 | 14.06 | 14.75 |
| South | 28.22 | 29.49 | 28.99 |
| Total (N) | 21,776 | 34,955 | 56,731 |

Source: Author's estimation based on NSSO survey, 75[th] round, 2017–18.

distribution of households by social classes shows that almost 43% of NCDs households are from other backward classes, 23% are from scheduled caste/tribe, and 34% are from other categories. Similarly, for non-NCDs households, it is 44%, 29%, and 27% for OBC, other categories, and SC/ST, respectively. Of those with hospitalisation cases, 96% NCDs households are drinking safe water. However, the figure is also same for non-NCDs households. NCDs households with 1–5 members have a higher rate of hospitalisation (66%) than those with 6–7 (23%) and 8+ (11%) members. Similarly, the hospitalisation rate is 71% for non-NCDs households having 1–5 members, 20% for 6–7 members, and 9% for 8+ members. The analysis of hospitalisation cases by wealth quintiles indicates that households with better economic status are being hospitalised more as compared to the households with lower economic status. At the regional level, more households are found to be hospitalised in the case of the southern region as compared to any other region.

## Out-of-pocket expenditure on hospitalisation by NCDs and Non-NCDs households

Fig 1 shows mean OOPE by type of disease (NCDs and non-NCDs) by type of healthcare facilities. The total expenditure by NCDs households is INR 35, 512, whereas it is INR 21, 214 for non-NCDs households. However, there is a huge difference in mean OOPE found between NCDs and non-NCDs households in public facilities, where it is INR 13,170 for NCDs households which is more than twice as compared to the non-NCDs households (INR 6,245).

The details of mean OOPE on hospitalisation by the public facility have been reported in Table 2. In the case of NCDs households, medical expenditure as a proportion of total expenditure is highest (0.80), followed by non-medical expenditure (0.13), and transport (0.06). Similarly, for non-NCDs households, it is 0.74, 0.17, and 0.08 for medical, non-medical, and transport expenditure respectively. Total expenditure on hospitalisation in the case of NCDs is higher (INR 22,808) for those households having more than two elder members as compared to the non-NCDs households (INR 10,795) having the same number of elderly. The rural-urban difference in total expenditure is more in the case of non-NCDs households as

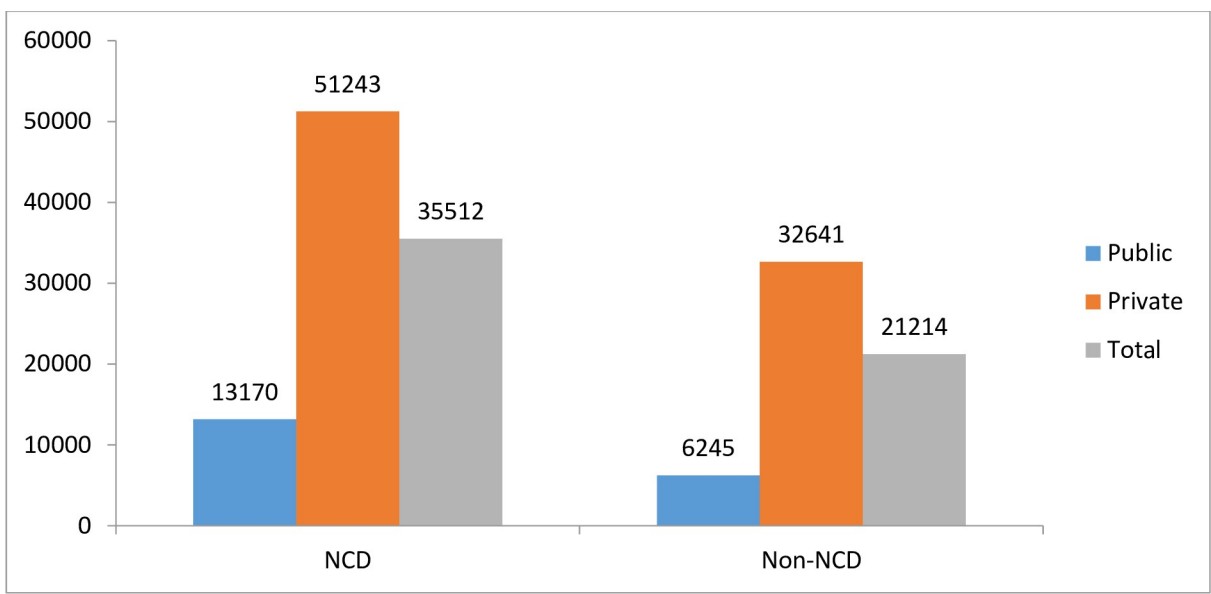

**Fig 1. Mean OOPE by type of disease and healthcare facility, 2017–18.**

**Table 2. Mean out-of-pocket expenditure by NCDs and non-NCDs households in public health care facility, NSSO survey 2017–18.**

| Household's Characteristic | Out-of-pocket expenditure in INR (US$) | | | | | | | |
|---|---|---|---|---|---|---|---|---|
| | NCDs households | | | | Non-NCDs households | | | |
| | Medical | Transport | Other non-medical | Total | Medical | Transport | Other non-medical | Total |
| Elderly member | | | | | | | | |
| No elderly | **8630 (133)** | **770 (12)** | **1614 (25)** | **11014 (170)** | **4066 (63)** | **495 (8)** | **1004 (15)** | **5564 (86)** |
| One elderly | **10186 (157)** | **957 (15)** | **1870 (29)** | **13013 (201)** | **5059 (78)** | **628 (10)** | **1194 (18)** | **6881 (106)** |
| 2+ elderly | **19539 (301)** | **1178 (18)** | **2092 (32)** | **22808 (352)** | **8638 (133)** | **669 (10)** | **1488 (23)** | **10795 (166)** |
| Residence | | | | | | | | |
| Rural | 10372 (160) | **936 (14)** | 1772 (27) | 13080 (202) | **4270 (66)** | 548 (8) | **1087 (17)** | **5905 (91)** |
| Urban | 10911 (168) | **751 (12)** | 1694 (26) | 13355 (206) | **5630 (87)** | 495 (8) | 1059 (16) | **7184 (111)** |
| Religion | | | | | | | | |
| Hindu | 10790 (166) | 858 (13) | 1769 (27) | 13417 (207) | 4657 (72) | **536 (8)** | **1102 (17)** | 6295 (97) |
| Muslim | 8244 (127) | 895 (14) | 1532 (24) | 10671 (165) | 4231 (65) | **507 (8)** | **936 (14)** | 5674 (87) |
| Others | 14072 (217) | 1120 (17) | 2082 (32) | 17274 (266) | 5595 (86) | **595 (9)** | **1190 (18)** | 7380 (114) |
| Caste | | | | | | | | |
| SC/ST | **9201 (142)** | **903 (14)** | 1723 (27) | **11828 (182)** | **3340 (51)** | **484 (7)** | 1055 (16) | **4879 (75)** |
| OBC | **9468 (146)** | **849 (13)** | 1915 (30) | **12232 (189)** | **4482 (69)** | **519 (8)** | 1117 (17) | **6118 (94)** |
| Others | **13304 (205)** | **881 (14)** | 1552 (24) | **15738 (243)** | **6660 (103)** | **627 (10)** | 1050 (16) | **8338 (129)** |
| Household size | | | | | | | | |
| 1–5 | 10653 (164) | 869 (13) | 1753 (27) | 13275 (205) | **4107 (63)** | **504 (8)** | **1021 (16)** | **5632 (87)** |
| 6–7 | 9858 (152) | 935 (14) | 1686 (26) | 12480 (192) | **5381 (83)** | **625 (10)** | **1185 (18)** | **7191 (111)** |
| 8+ | 11437 (176) | 777 (12) | 1845 (28) | 14059 (217) | **8418 (130)** | **612 (9)** | **1438 (22)** | **10468 (161)** |
| Cooking fuel | | | | | | | | |
| Firewood | **9123 (141)** | **1008 (16)** | 1807 (28) | **11939 (184)** | **3982 (61)** | 566 (9) | 1033 (16) | **5581 (86)** |
| LPG/Other gas | **11963 (184)** | **753 (12)** | 1692 (26) | **14408 (222)** | **4972 (77)** | 503 (8) | 1114 (17) | **6589 (102)** |
| Other | **7991 (123)** | **1002 (15)** | 1791 (28) | **10783 (166)** | **7406 (114)** | 527 (8) | 1183 (18) | **9116 (141)** |
| Drinking water | | | | | | | | |
| Safe | 10234 (158) | **836 (13)** | **1715 (26)** | 12786 (197) | 4622 (71) | **530 (8)** | **1067 (16)** | 6220 (96) |
| Unsafe | 17456 (269) | **1739 (27)** | **2431 (37)** | 21626 (333) | 4905 (76) | **632 (10)** | **1421 (22)** | 6958 (107) |
| Type of latrine | | | | | | | | |
| Flush/pit | **11385 (176)** | 876 (14) | 1690 (26) | **13950 (215)** | 4706 (73) | 539 (8) | **1043 (16)** | 6288 (97) |
| Open space/others | **7387 (114)** | 874 (13) | 1961 (30) | **10222 (158)** | 4415 (68) | 520 (8) | **1186 (18)** | 6121 (94) |
| Wealth quintile | | | | | | | | |
| Poorest | **7628 (118)** | **681 (11)** | **1413 (22)** | **9722 (150)** | 4266 (66) | 507 (8) | **1001 (15)** | 5774 (89) |
| Poorer | **7003 (108)** | **773 (12)** | **1428 (22)** | **9204 (142)** | 5275 (81) | 474 (7) | **1022 (16)** | 6770 (104) |
| Middle | **9075 (140)** | **786 (12)** | **1694 (26)** | **11554 (178)** | 4520 (70) | 519 (8) | **1062 (16)** | 6101 (94) |
| Richer | **10563 (163)** | **984 (15)** | **2029 (31)** | **13577 (209)** | 4510 (70) | 582 (9) | **1096 (17)** | 6188 (95) |
| Richest | **17339 (267)** | **1106 (17)** | **2096 (32)** | **20541 (317)** | 4627 (71) | 575 (9) | **1197 (18)** | 6399 (99) |
| Region | | | | | | | | |
| North | **15496 (239)** | **1164 (18)** | **1892 (29)** | **18552 (286)** | **5970 (92)** | **657 (10)** | **1174 (18)** | **7801 (120)** |
| Central | **9622 (148)** | **656 (10)** | **1465 (23)** | **11743 (181)** | **6139 (95)** | **405 (6)** | **987 (15)** | **7531 (116)** |
| East | **11900 (183)** | **931 (14)** | **1477 (23)** | **14308 (221)** | **4340 (67)** | **557 (9)** | **928 (14)** | **5826 (90)** |
| Northeast | **7239 (112)** | **750 (12)** | **1558 (24)** | **9546 (147)** | **4554 (70)** | **610 (9)** | **1018 (16)** | **6182 (95)** |
| West | **9055 (140)** | **594 (9)** | **1235 (19)** | **10884 (168)** | **5003 (77)** | **475 (7)** | **745 (11)** | **6224 (96)** |
| South | **7331 (113)** | **904 (14)** | **2419 (37)** | **10653 (164)** | **3111 (48)** | **515 (8)** | **1388 (21)** | **5014 (77)** |
| Total (N) | 10549 (163) | 875 (13) | 1747 (27) | 13170 (203) | 4632 (71) | 534 (8) | 1079 (17) | 6245 (96) |

INR: Indian rupees and values in bracket are in terms of US dollar (US$) as per average exchange rate in 2017–18 (64.86; www.rbi.org.in).

Note:- Results in bold are significant at 5% level as tested by ANOVA.

compared to the NCDs households. In comparison to the Muslim religion, households from the Hindu religious category are spending more on hospitalisation for the treatment of both NCDs (INR 13,417) and non-NCDs (INR 6,295). Households from SC/ST categories are spending INR 11,828 due to hospitalisation associated with NCDs, whereas it is INR 4,879 for non-NCDs hospitalisation. The mean OOPE is more for households having more than 8 members as compared with the households having 1–7 members, both in the case of NCDs and non-NCDs. People from NCDs households who are drinking unsafe water have higher medical expenses (INR 17,456) when compared to non-NCD households (INR 4,905). The economic status of households significantly influenced spending on hospitalisation. It clearly shows that the richest households are spending more as compared to all other income groups. However, the spending is more in the case of NCDs households (INR 20,541) than non-NCDs households (INR 6,399).

Table 3 presents the mean OOPE by private facilities. It shows that the total expenditure on hospitalisation in the case of NCDs households is INR 51,243, whereas it is INR 32,641 for non-NCDs households. Medical expenditure constitutes a larger share of total expenditure experienced by both NCDs and non-NCDs households. Particularly, the proportion of total medical expenditure incurred on medicines (0.40) and diagnostics (0.15) is troublesome for households with NCDs, treated in public hospitals. Similarly, the proportion of total medical expenditure incurred on diagnostic tests is higher (0.10) for NCDs households availing private facilities as compared to the non-NCDs households (0.09) (S1 Appendix).

## Catastrophic health expenditure on hospitalisation by NCDs and non-NCDs households

CHE on hospitalisation by household's socio-economic characteristics has been presented in Table 4. At the public facility, 27.68% of NCDs households are exposed to CHE due to hospitalisation and 14.59% of non-NCDs households are experiencing the same. The incidence of CHE is even more in the case of private facilities, where it is 72.09% (3 times more than public hospitals) for NCDs households and 55.85% for non-NCDs households. Households where the number of elderly is higher incurred a greater burden of CHE (74.75%) due to hospitalisation associated with NCDs at private hospitals. The level of CHE is higher for rural NCDs households as compared to the rural non-NCDs households in both public and private facilities; however, the difference is more in public facilities. In private facilities, CHE for NCDs households is highest (73.05%) for those who belong to the Hindu religion as compared to Muslims and others. Similarly, NCDs households from SC/ST category encounter more CHE (73.58%) than non-NCDs households (58.56%) in private facilities.

With the increase in household size from 1–5 members to 6–7 and 8+ members, the level of CHE decreases. In the case of non-NCDs households, CHE is more for those households who are using firewood (62.06%), compared with those who are using LPG and other gas (53.82%) in private facilities. The extent of CHE is more for those households who are using unsafe drinking water and open space for defecation. In private facilities, the incidence of CHE is 67.01% for the poorest wealth quintile and 49.83% for the richest quintile in the case of non-NCDs households, while in the case of NCDs households it is 82.58% and 68.83% for the poorest and richest category respectively. CHE in the northern region is 28.57% for NCDs households, which is 2 times more than the CHE experienced by the non-NCDs households, in public facilities.

## Results from generalised linear model

Results from Table 5 indicate the socio-economic determinants of CHE among the households in India. It shows that households with more elderly members are incurring more CHE as

**Table 3. Mean out-of-pocket expenditure by NCDs and non-NCDs households in private health care facilities, NSSO survey 2017–18.**

| Household's Characteristic | Out-of-pocket expenditure in INR (US$) | | | | | | | |
| --- | --- | --- | --- | --- | --- | --- | --- | --- |
| | NCDs households | | | | Non-NCDs households | | | |
| | Medical | Transport | Other non-medical | Total | Medical | Transport | Other non-medical | Total |
| Elderly member | | | | | | | | |
| No elderly | **40041 (617)** | **1115 (17)** | **2300 (35)** | **43456 (670)** | **26472 (408)** | **813 (13)** | **1790 (28)** | **29074 (448)** |
| One elderly | **50805 (783)** | **1365 (21)** | **2816 (43)** | **54986 (848)** | **35659 (550)** | **946 (15)** | **2569 (40)** | **39174 (604)** |
| 2+ elderly | **66587 (1027)** | **1460 (23)** | **2950 (45)** | **70997 (1095)** | **38693 (597)** | **1028 (16)** | **2374 (37)** | **42095 (649)** |
| Residence | | | | | | | | |
| Rural | **38798 (598)** | 1313 (20) | 2612 (40) | **42723 (659)** | **26644 (411)** | 954 (15) | 2193 (34) | **29791 (459)** |
| Urban | **59762 (921)** | 1134 (17) | 2454 (38) | **63350 (977)** | **34741 (536)** | 722 (11) | 1735 (27) | **37199 (574)** |
| Religion | | | | | | | | |
| Hindu | **47532 (733)** | 1263 (19) | 2553 (39) | **51348 (792)** | 30202 (466) | 876 (14) | 2088 (32) | 33167 (511) |
| Muslim | **41131 (634)** | 1179 (18) | 2408 (37) | **44718 (689)** | 26701 (412) | 839 (13) | 1662 (26) | 29202 (450) |
| Others | **57263 (883)** | 1106 (17) | 2712 (42) | **61081 (942)** | 29940 (462) | 772 (12) | 1796 (28) | 32508 (501) |
| Caste | | | | | | | | |
| SC/ST | **38999 (601)** | **1114 (17)** | 2388 (37) | **42501 (655)** | **29260 (451)** | **831 (13)** | 2613 (40) | **32704 (504)** |
| OBC | **39931 (616)** | **1199 (18)** | 2401 (37) | **43531 (671)** | **26030 (401)** | **865 (13)** | 1802 (28) | **28697 (442)** |
| Others | **60994 (940)** | **1352 (21)** | 2806 (43) | **65152 (1004)** | **35303 (544)** | **885 (14)** | 1942 (30) | **38130 (588)** |
| Household size | | | | | | | | |
| 1–5 | 47672 (735) | 1206 (19) | 2502 (39) | 51379 (792) | 29931 (461) | **830 (13)** | 2066 (32) | 32827 (506) |
| 6–7 | 41629 (642) | 1228 (19) | 2415 (37) | 45272 (698) | 27899 (430) | **932 (14)** | 1930 (30) | 30761 (474) |
| 8+ | 57831 (892) | 1463 (23) | 3076 (47) | 62371 (962) | 32523 (501) | **960 (15)** | 1864 (29) | 35347 (545) |
| Cooking fuel | | | | | | | | |
| Firewood | **37311 (575)** | 1367 (21) | 2686 (41) | **41364 (638)** | **25935 (400)** | **972 (15)** | **1817 (28)** | **28725 (443)** |
| LPG/Other gas | **52189 (805)** | 1196 (18) | 2482 (38) | **55867 (861)** | **31713 (489)** | **828 (13)** | **2145 (33)** | **34686 (535)** |
| Other | **39581 (610)** | 1127 (17) | 2641 (41) | **43349 (668)** | **23856 (368)** | **774 (12)** | **1336 (21)** | **25967 (400)** |
| Drinking water | | | | | | | | |
| Safe | 47453 (732) | **1227 (19)** | 2547 (39) | 51228 (790) | 29834 (460) | **868 (13)** | **2026 (31)** | 32728 (505) |
| Unsafe | 47589 (734) | **1641 (25)** | 2522 (39) | 51752 (798) | 27668 (427) | **771 (12)** | **1769 (27)** | 30208 (466) |
| Type of latrine | | | | | | | | |
| Flush/pit | **50688 (781)** | **1261 (19)** | **2586 (40)** | **54534 (841)** | **31386 (484)** | 851 (13) | 2073 (32) | **34310 (529)** |
| Open space/others | **30209 (466)** | **1125 (17)** | **2337 (36)** | **33671 (519)** | **22195 (342)** | 929 (14) | 1756 (27) | **24880 (384)** |
| Wealth quintile | | | | | | | | |
| Poorest | **28067 (433)** | 1000 (15) | **1826 (28)** | **30894 (476)** | 24202 (373) | 854 (13) | 1529 (24) | 26585 (410) |
| Poorer | **39741 (613)** | 1159 (18) | **2567 (40)** | **43467 (670)** | 28261 (436) | 814 (13) | 2495 (38) | 31570 (487) |
| Middle | **44260 (682)** | 1261 (19) | **2387 (37)** | **47909 (739)** | 29211 (450) | 878 (14) | 1795 (28) | 31884 (492) |
| Richer | **43423 (669)** | 1247 (19) | **2334 (36)** | **47004 (725)** | 27073 (417) | 831 (13) | 1732 (27) | 29636 (457) |
| Richest | **62718 (967)** | 1355 (21) | **3024 (47)** | **67097 (1034)** | 34800 (537) | 909 (14) | 2304 (36) | 38013 (586) |
| Region | | | | | | | | |
| North | **55543 (856)** | **1420 (22)** | **2715 (42)** | **59678 (920)** | 32252 (497) | 977 (15) | 1839 (28) | 35068 (541) |
| Central | **46149 (712)** | **1240 (19)** | **2613 (40)** | **50002 (771)** | 33232 (512) | 1052 (16) | 1927 (30) | 36211 (558) |
| East | **44880 (692)** | **1508 (23)** | **2821 (43)** | **49210 (759)** | 32402 (500) | 1075 (17) | 2539 (39) | 36016 (555) |
| Northeast | **55171 (851)** | **3325 (51)** | **4093 (63)** | **62589 (965)** | 31933 (492) | 1680 (26) | 3404 (52) | 37016 (571) |
| West | **47239 (728)** | **1050 (16)** | **1866 (29)** | **50155 (773)** | **27784 (428)** | **618 (10)** | **1306 (20)** | **29708 (458)** |
| South | **46363 (715)** | **1115 (17)** | **2705 (42)** | **50183 (774)** | **26458 (408)** | **731 (11)** | **2312 (36)** | **29501 (455)** |
| Total (N) | 47457 (732) | 1239 (19) | 2547 (39) | 51243 (790) | 29759 (459) | 865 (13) | 2017 (31) | 32641 (503) |

INR: Indian rupees and values in bracket are in terms of US dollar (US$) as per average exchange rate in 2017–18 (64.86; www.rbi.org.in).

Note:- Results in bold are significant at 5% level as tested by ANOVA.

**Table 4. Percentage of households experiencing CHE due to hospitalisation by type of facility, NSSO survey 2017–18.**

| Household's Characteristic | Public | | Private | |
|---|---|---|---|---|
| | NCDs | Non-NCDs | NCDs | Non-NCDs |
| Elderly member | | | | |
| No elderly | 25.70 | 13.14 | 70.75 | 54.06 |
| One elderly | 30.24 | 18.03 | 73.10 | 59.88 |
| 2+ elderly | 31.67 | 20.90 | 74.75 | 60.10 |
| Residence | | | | |
| Rural | 30.24 | 16.66 | 76.23 | 61.46 |
| Urban | 23.85 | 10.66 | 68.27 | 50.27 |
| Religion | | | | |
| Hindu | 28.08 | 15.24 | 73.05 | 56.69 |
| Muslim | 26.25 | 12.29 | 68.54 | 53.17 |
| Others | 27.16 | 13.76 | 68.06 | 51.53 |
| Caste | | | | |
| SC/ST | 28.24 | 14.54 | 73.58 | 58.56 |
| OBC | 28.19 | 15.08 | 71.97 | 56.00 |
| Others | 26.45 | 13.99 | 71.47 | 53.98 |
| Household size | | | | |
| 1–5 | 29.75 | 15.19 | 75.24 | 58.54 |
| 6–7 | 24.38 | 12.92 | 68.29 | 51.38 |
| 8+ | 20.75 | 13.25 | 60.64 | 46.14 |
| Cooking fuel | | | | |
| Firewood | 32.56 | 17.32 | 75.55 | 62.06 |
| LPG/Other gas | 24.54 | 12.04 | 70.99 | 53.82 |
| Other | 26.84 | 19.59 | 72.98 | 57.81 |
| Drinking water | | | | |
| Safe | 27.28 | 14.37 | 72.11 | 55.66 |
| Unsafe | 34.77 | 18.67 | 71.66 | 61.80 |
| Type of latrine | | | | |
| Flush/pit | 26.78 | 13.17 | 71.20 | 54.41 |
| Open space/others | 32.57 | 20.85 | 78.47 | 64.48 |
| Wealth quintile | | | | |
| Poorest | 35.86 | 22.76 | 82.58 | 67.01 |
| Poorer | 29.60 | 15.06 | 75.20 | 60.73 |
| Middle | 26.27 | 14.02 | 73.56 | 57.19 |
| Richer | 24.51 | 11.95 | 67.95 | 53.54 |
| Richest | 23.90 | 10.33 | 68.83 | 49.83 |
| Region | | | | |
| North | 28.89 | 15.58 | 67.91 | 50.69 |
| Central | 29.52 | 16.90 | 74.93 | 63.27 |
| East | 32.50 | 18.77 | 74.09 | 59.61 |
| Northeast | 28.57 | 13.42 | 71.06 | 58.45 |
| West | 18.04 | 10.03 | 67.64 | 48.27 |
| South | 22.57 | 10.15 | 74.66 | 55.90 |
| Total (%) | 27.68 | 14.59 | 72.09 | 55.85 |

Source: Author's estimation based on NSSO survey, 75th round, 2017–18.

**Table 5. Socio-economic factors associated with CHE due to hospitalisation by type of facility, NSSO survey 2017–18.**

| Household's characteristics | Public | | Private | |
|---|---|---|---|---|
| | NCDs | Non-NCDs | NCDs | Non-NCDs |
| No elderly (Ref.) | | | | |
| One elderly | 0.290* | 0.530* | 0.040* | 0.127* |
| 2+ elderly | 0.426* | 0.422* | 0.130* | 0.217* |
| Rural (Ref.) | | | | |
| Urban | -0.402* | -0.458* | -0.143* | -0.250* |
| Hindu (Ref.) | | | | |
| Muslim | 0.065 | -0.291* | -0.081* | -0.167* |
| Others | 0.511* | 0.344* | -0.100* | -0.161* |
| SC/ST (Ref.) | | | | |
| OBC | 0.005 | 0.415* | -0.058* | 0.009 |
| Others | -0.061 | 0.465* | 0.029 | 0.060* |
| 1–5 (Ref.) | | | | |
| 6–7 | -0.609* | -0.402* | -0.175* | -0.306* |
| 8+ | -0.518* | -0.396* | -0.303* | -0.477* |
| Firewood (Ref.) | | | | |
| LPG/Other gas | -0.059 | -0.278* | 0.005 | 0.012 |
| Other | -0.140 | 0.208** | -0.023 | -0.015 |
| Safe (Ref.) | | | | |
| Unsafe | 0.351* | 0.089 | -0.094* | -0.008 |
| Flush/pit (Ref.) | | | | |
| Open space/others | -0.073 | 0.097*** | 0.014 | -0.021 |
| Poorest (Ref.) | | | | |
| Poorer | -0.335* | -0.369* | -0.127* | -0.127* |
| Middle | -0.565* | -0.667* | -0.200* | -0.257* |
| Richer | -0.593* | -0.848* | -0.302* | -0.366* |
| Richest | -0.656* | -1.059* | -0.314* | -0.429* |
| North (Ref.) | | | | |
| Central | -0.018 | -0.007 | 0.034 | 0.188* |
| East | -0.015 | 0.008 | 0.013 | 0.129* |
| Northeast | -0.272** | -0.132 | -0.030 | 0.089 |
| West | -0.425* | -0.208** | -0.055** | -0.127* |
| South | -0.312* | -0.212* | 0.006 | 0.000 |

Source: Author's estimation based on NSSO survey, 75th round, 2017–18.

*significant at 1% level

**significant at 5% level.

compared to the households with no elderly in both private and public facilities. At the public facility, CHE is less in the case of NCDs households living in the urban area ($\beta$ = -0.402*) in comparison to rural areas. Analysis by religious category shows that NCDs households from other religious categories are experiencing more catastrophic spending ($\beta$ = 0.511*) than households with Hindu religion at the public facility. However, NCDs households availing private facility shows that other religious categories (Christianity, Jainism, Sikhism, Buddhism) are experiencing less CHE ($\beta$ = -0.100*) as compared to the Hindu religious category.

Similarly, the incidence of CHE in the case of non-NCDs households is significantly higher for the OBC category than SC/ST in public facilities. Size of households is another important

determinant of CHE that indicates the incidence of CHE is significantly lower for households having 6–7 and 8+ members as compared to the reference category. Similarly, sources of cooking indicate that in public facilities non-NCDs related CHE is less ($\beta$ = -0.278*) for those households who are using LPG and other gas as compared with those who are using firewood. In the public facilities, NCDs attributable CHE is more ($\beta$ = 0.351*) for those households who are using unsafe water as compared to the households using safe water. The economic status of the household is significantly associated with CHE, which shows that CHE is lower for the households with better economic status as compared to the households with poor economic status. Similarly, the incidence of CHE is significantly lesser for the NCDs households from the Northeast (-0.272**), West (-0.425*), and Southern region (-0.312*) when compared to the reference category in the public facilities.

## Discussion

The primary goal of the 2011 United Nations high-level meeting on NCDs is to protect people from premature death caused by NCDs like stroke, diabetes, cancer, and respiratory diseases. To achieve this target, the WHO Global Action Plan on the prevention and treatment of NCDs puts greater emphasis on ensuring affordable access to early diagnosis and treatment for those with NCDs. Our result shows that we are far from achieving this goal because the households that have a member hospitalised due to NCDs are more vulnerable to catastrophic spending as compared to someone hospitalised due to other communicable or infectious diseases.

Socio-economic characteristics of sample households demonstrate that out of 56,731 households with hospitalisation cases, 21,776 have been hospitalised due to NCDs, and 34,955 households have been hospitalised due to some injuries/communicable/ infectious diseases, other than NCDs. Majority of household (both NCDs and non-NCDs) belongs to other backward class, Hindu religion, and residing in the rural area. There are fewer households from poor economic status have been hospitalised as compared to the richer households. Here the possible reason might be that poor people are not going for seeking care due to cost constraints [41, 42]. Notable regional variation in hospitalisation cases is found in the present study with the highest number of hospitalisation cases reported from the southern region and lowest from the northeast. Literature that supports this finding is that southern states are more affluent in terms of higher per capita income, which may cause them to seek more healthcare, as compared to other states [43].

NCDs induced households are experiencing more OOPE than households with non-NCDs. As per our findings, the economic burden of NCDs (measured in terms of OOPE) in public hospitals is more than twice for NCDs affected households as compared to non-NCDs households. In line with our findings, previous studies also found that NCDs affected households are spending comparatively higher OOPE than households with non-NCDs [44–46]. The share of medical expenditure is highest in the total expenditure followed by other non-medical and transport expenditure. A substantial proportion of medical expenditure is for medication, diagnosis, and other medical expenditures like physiotherapy, blood, oxygen, etc. Consistent with other studies, OOPE is much more in private facilities, relative to public facilities [47–48].

The study further reveals that the burden of OOPE is disproportionately distributed among the different subgroups of the population. The burden is more among the households having more than two elderly members in both public and private facilities. This may be because the elderly are more prone to multiple health conditions than the younger adults. Similar findings are found in many other studies [49–51]. In public health centers, the rural-urban difference in mean OOPE is found to be very less among NCDs households, but the difference is more

while considering non-NCDs households. However, in both cases, the brunt of OOPE is more among urban people as compared to rural counterparts, likely due to the higher cost of treatment in urban areas than rural. Analysis by religious affiliation shows that households that belong to the Hindu religion are spending more on hospitalisation as compared to Muslim households. As evidenced by the previous literature, Muslims have higher poverty rates and lower education levels than Hindus, which may explain why they spend less money on hospitalisation [33, 52]. Hospitalisation is found to be varied by different social groups, indicating that mean OOPE (in both private and public facilities) is lower for SC/ST and OBC households than the households of other castes. This finding demonstrates that even in the twenty-first century, despite all medical advances and institutional changes, social institutions continue to have a considerable impact on household healthcare seeking behaviour and expenditure patterns [53].

In public facilities, households using unsafe drinking water are reporting a higher burden of OOPE expenditure compared to the households using safe drinking water. This is because the chances of infection are more in the case of the former than in the latter. As documented by previous studies [54, 55], OOPE on hospitalisation is found to be directly related to the economic status of the households which indicates that households with lower economic status are spending less compared to households with higher economic status, reflecting the ability to pay principle of paying for healthcare.

Regarding the economic burden of NCDs in terms of CHE, our result shows that overall, 27.68% of NCDs affected households and 14.59% of non-NCDs households are experiencing CHE in public facilities, whereas, in the private facilities, it is 72.09% and 55.85% for NCDs and non-NCDs households respectively. A higher incidence of CHE is found among the households from the lower-income category compared to the households from the higher-income group which indicates that NCDs disproportionately affect poor households, thus increasing inequalities. The burden of CHE is hefty for rural households as compared to their urban counterparts. This may be because in rural areas a larger proportion of households are already concentrated around the poverty line, as a result of which a smaller proportion of OOPE leads to catastrophic spending.

The result from generalised linear regression model shows a significant relationship between CHE with residence, caste, religion, household size, economic status of households, and households having more elderly members. The intensity of CHE is more for the households who are poor, drinking unsafe water, using firewood as cooking fuel, and household size of 1–5 members.

## Conclusion

Our study provides evidence of the economic burden of NCDs faced by households in India and associated socio-economic factors with it. Compared to the non-NCDs households, the healthcare burden in terms of OOPE is higher for the NCDs affected households, particularly for those who are seeking care in private facilities than in public healthcare facilities. Therefore, our analysis shows that despite the effort made by the government of India in introducing various social insurance schemes, a notable proportion of Indian households are still facing higher CHE due to NCDs. Based on our findings, it can be said that India is far from achieving financial risk protection for the people with NCDs, in the context of SDGs. There is an urgent need for government to make affordable health insurance policies available to the economically weaker sections to protect them from the catastrophic cost of treating NCDs. Particularly, a customised disease-specific health insurance package should be introduced by the government of India in both the public and private facilities, as well as raise public awareness about the

availability of the same. As NCDs have a disproportionate economic impact on poor households, redistributive measures such as income taxation, subsidies for healthy substitutes, and intervention targeting vulnerable populations may need to be considered for population-based strategies to combat NCDs diseases.

## Limitations of the study

The current research has some limitations. First, the cross-sectional nature of data enables us to investigate only the short-term impact of NCDs on household OOPE payments. Secondly, since NSSO data are self-reported, there could be a chance of over- or under-estimation of results. Third, it deals with the consumption expenditure of households and ignores the income aspect associated with healthcare spending. Fourth, this study was unable to account for any price differentials or the cost impact of healthcare, particularly for those who had not sought any medical attention.

## Supporting information

**S1 Appendix. Out-of-pocket expenditure as a proportion of total medical expenditure, NSSO survey 2017–18.**
(DOCX)

## Acknowledgments

The authors are grateful to the Department of Humanities and Social Sciences, National Institute of Technology (NIT), Rourkela for their support and encouragement, which has helped in improving this paper.

## Author Contributions

**Conceptualization:** Sasmita Behera.

**Supervision:** Jalandhar Pradhan.

**Writing – original draft:** Sasmita Behera.

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
