## [Decision Letter · Decision Letter 0]

6 Sep 2021

PONE-D-21-18147Uneven economic burden of non-communicable diseases among Indian households: A comparative analysisPLOS ONE

Dear Dr. BEHERA,

Thank you for submitting your manuscript to PLOS ONE. After careful consideration, we feel that it has merit but does not fully meet PLOS ONE’s publication criteria as it currently stands. Therefore, we invite you to submit a revised version of the manuscript that addresses the points raised during the review process. The revised version should address all comments. Please submit your revised manuscript by Oct 21 2021 11:59PM. If you will need more time than this to complete your revisions, please reply to this message or contact the journal office at plosone@plos.org. Please include the following items when submitting your revised manuscript:A rebuttal letter that responds to each point raised by the academic editor and reviewer(s). You should upload this letter as a separate file labeled 'Response to Reviewers'.A marked-up copy of your manuscript that highlights changes made to the original version. You should upload this as a separate file labeled 'Revised Manuscript with Track Changes'.An unmarked version of your revised paper without tracked changes. You should upload this as a separate file labeled 'Manuscript'.

We look forward to receiving your revised manuscript.

Kind regards,

Petri Böckerman

Academic Editor

PLOS ONE

Journal Requirements:

Whilst you may use any professional scientific editing service of your choice, PLOS has partnered with both American Journal Experts (AJE) and Editage to provide discounted services to PLOS authors. Both organizations have experience helping authors meet PLOS guidelines and can provide language editing, translation, manuscript formatting, and figure formatting to ensure your manuscript meets our submission guidelines. To take advantage of our partnership with AJE, visit the AJE website (http://aje.com/go/plos) for a 15% discount off AJE services. To take advantage of our partnership with Editage, visit the Editage website (www.editage.com) and enter referral code PLOSEDIT for a 15% discount off Editage services.  If the PLOS editorial team finds any language issues in text that either AJE or Editage has edited, the service provider will re-edit the text for free.

5. Please remove your figures from within your manuscript file, leaving only the individual TIFF/EPS image files, uploaded separately.  These will be automatically included in the reviewers’ PDF.

Reviewers' comments:

Reviewer's Responses to Questions

**Comments to the Author**

1. Is the manuscript technically sound, and do the data support the conclusions?

Reviewer #1: Partly

Reviewer #2: Yes

2. Has the statistical analysis been performed appropriately and rigorously? 

Reviewer #1: No

Reviewer #2: N/A

3. Have the authors made all data underlying the findings in their manuscript fully available?

Reviewer #1: Yes

Reviewer #2: Yes

4. Is the manuscript presented in an intelligible fashion and written in standard English?

Reviewer #1: No

Reviewer #2: Yes

5. Review Comments to the Author

Reviewer #1: Thanks for providing me the opportunity to review this manuscript. The authors analyze the Indian national survey NSSO 75th round. The analysis has value in terms understanding the economic burden of NCDs in India. This research could provide reflections on how the recent healthcare reforms in lines with India’s UHC aspirations has succeeded/failed to produce coveted effects. Presently the analysis and overall manuscript stand week. There are several lacks related to methodological description, presentation of descriptive results, theoretical basis of choice of variables in regression, focused discussion, policy implications etc. This manuscript can benefit from major revision. The specific comments for each section are given below.

Abstract

1. The term ‘lower and middle income countries’ is incorrect, it is not clear whether the authors are referring to ‘low and middle income countries’ or lower-middle income countries’ please revise accordingly.

2. Please provide the acronym-NCDs at its first use.

3. Please elaborate the methods used.

4. Results: please mention if you are reporting mean expenditures

5. Results: please replace ‘2 times more’ with ‘more than twice’

6. Results: The text ‘Particularly, expenditures on medication and diagnosis are troublesome for households with NCDs.’ seems generic. It would be good if authors can provide proportion of total OOP expenditure incurred on medicines and diagnostics.

7. For results of generalised linear regression model please provide odds ratio and p-values.

8. Presently, the conclusion statement is crude. It needs to be specific with expansion.

Introduction

1. General comments: There are multiple grammatical errors in the text. I have attempted to point out many of these mistakes, but a thorough copy-edit is required to catch all of these errors.

Additionally, the introduction is unnecessarily lengthy owing to repetition of concepts. The structure of the introduction needs to be reworked to ensure that distinct ideas are grouped together into paragraphs and that concepts flow to build a context across paragraphs.

Specific comments:

1. Page 2 line 3, as explained above please replace the term ‘lower and middle income countries’

2. Page 2 line 3 please replace the word ‘kills’ with ‘kill’

3. Page 2 line 4 please pluralize the word ‘death’

4. Please be consistent in the use of the acronym for non-communicable diseases. There is an inconsistency in the use of the full form and the acronym. Additionally at some places ‘NCD’ is used and at others it is ‘NCDs’.

5. Page 2 line 9 please pluralize the verb ‘trap’

6. Page 2 line 10 please replace the word like with ‘such as’

7. Page 2 line 11 again the authors have used the full form instead of the abbreviation for non-communicable diseases. It is advised to give the full form at first-in-text reference along with the acronym in parenthesis and then use the acronym in the subsequent references.

8. Page 2 line 13 please revise the sentence ‘the economic cost of NCDs..’ for it to be grammatically correct. Additionally please specify ‘…macroeconomic effect’ for whom?

9. Page 2 line 14 please replace the word ‘force’ with ‘workforce’

10. Page 3 lines 1-2, there is incorrect use of prepositions, please revise accordingly.

11. Page 3 lines 4-5, please do not capitalize the words succeeding colon and semi colon.

12. Page 3 line 15 please pluralize the word ‘death’

13. Page 3 line 15 please add ‘the’ before the word ‘..death are’

14. Page 3 Please do not provide the full form for DALYs again. Also look out for similar errors for other acronyms (UHC, SDG) used in the manuscript.

15. Page 3, the authors provide DALYs from 1990-2016, it is advised to provide more recent figures using GBD study 2019.

16. Page 3 please provide a reference for the sentence ‘OOPE constitutes 58.7% of total health expenditure in India’

17. Page 3 the figures cited by the authors ‘23 to 32 million people below the poverty line’ does not refer to impoverishment due to NCDs. Furthermore, the reference cited uses data from 1999-2000 which is more than two decades old. The authors are advised to accordingly revise these facts.

18. Page 3 please pluralize the word ‘policymaker’

19. Page 3 it is not clear how understanding of OOP expenditure would provide information on prevention and control of NCDs. Additionally, how would this reduce financial hardship. These lines need additional clarifications.

20. Page 4 contrary to the authors’ claim there are multiple studies assessing the OOP expenditure for NCDs apart from the two cited. Please provide a strong justification of the novelty of your study and how would it contribute to existing literature.

METHODOLOGY

General comments: The authors have analysed nationally representative NSSO data and provide useful descriptive results. The findings are important but empirically obvious. Furthermore, the depth of analysis is thin, the authors do not justify the methodology for computation of catastrophic health expenditures and the theoretical basis of selection of independent variables for their regression model.

Specific comments:

1. The authors do not provide a clear justification on how have they grouped the 15 broad disease categories into NCDs and non-NCDs. For instance respiratory ailments can be either NCDs or infectious diseases.

2. Page 5: under variables section it is not clear why the authors choose these particular variables in this analysis. For instance, why do the authors think religion is an important variable impacting your outcome variables-OOP expenditure due to NCDs.

Similarly, when you have already included wealth quintile as an independent variable why do you include type of ‘latrine’ which is a component of estimating proxy of economic status of household.

3. Page 6, the authors state that they use world bank criteria to measure OOPE. It is not clear what is meant by this since OOPE is already reported in NSSO data.

4. The authors do not provide a justification of calculating CHE based on 10% of total income. Previous literature reports that using consumption expenditure is a better indicator for calculating CHE in comparison to income. Further a range of thresholds is used 10%, 25%, 30% and 40%, please justify the selection of 10% cut-off.

5. Page 6 para 2 there is repetition of what is given in the preceding text.

6. Page 6 para 3, the authors provide theoretical description of a generalized linear model which may not be needed. Instead as mentioned previously please consider adding details on the reason for selection of variables in your model. Further, there is no mention about how multicollinearity issue was assessed and addressed?

Results

1. For results ‘Households hospitalised due to NCDs have more elderly members as compared to the non-NCD’ please mention corresponding figures-what proportion

2. ‘Nearly 80% of households, both NCDs and non-NCDs’ this is obvious depending on overall NSSO sample and India’ demographic statistics where majority of the population is Hindu.

3. Page 7, lines 4-5 please the authors report results for social class for NCDs households, please report corresponding results for Non-NCDs households as well.

4. Page 7 lines 6 the authors report combined results for NCD and non-NCDs households, please report the findings separately.

5. Also reporting results for all categories is redundant for example the authors state 96% drink safe water so it is not necessary to state that 4% are drinking unsafe water. It is suggested to report major findings only.

6. Please provide corresponding numbers ‘More number of hospitalisation cases

are found in the case of households having 1-5 members as compared to 6-7 and 8+ members’ and also how does it differ in NCDs versus non-NCDs households.

7. Page 7 in the sentence ‘As expected, analysis of hospitalisation.’, the authors should refrain from making speculations.

8. Overall, please use a standardized pattern for reporting descriptive results. For some variables the authors provide compare results for both NCDs and non-NCDs households and for others overall results for the complete sample are provided.

9. Page 9 lines 1-5, please provide results as proportion of total expenditure rather than absolute numbers.

10. The authors report Hindus are spending more than Muslims for NCDs is this difference statistically significant?

11. There is an inconsistency in reporting results for OOP expenditure, for some variables the authors make an intra-variable comparison (Hindus versus Muslims) for others the authors compare differences for NCDs vs non-NCDs households.

12. Table 2 and 3 please reconsider the title of the table ‘Difference in mean out-of-pocket expenditure’, the authors do not report the difference in these tables.

13. The authors report CHE is less for others categories for public hospitals and more for private hospitals with Hindus as reference category. Have the authors controlled for confounders for this analysis. Also in your discussion section please provide plausible reasons why this difference is observed.

14. Why is the role of gender not explored for OOPE and CHE. Previous literature has identified gender as one of the major independent variables affecting OOP and CHE

Discussion

General comments: The discussion section largely repeats the results of analysis and the authors do not comprehensively discuss the reasons for observing specific findings.

Specific comments

1. Page 18 line 1, the findings differ from what has been reported in results section and Table 1

2. Page 18 sentence ‘poor people are not going for seeking care due to cost constraints’, please reference this appropriately.

3. Page 18 sentence ‘highest number of hospitalisation cases reported from the southern region and lowest from the northeast’. Please discuss the basis of this finding.

4. ‘In public health centres, the rural-urban difference in mean OOPE is found to be very less among NCD households, however, this difference is more while considering non-NCD households. Why? Please discuss.

5. ‘households that belong to the Hindu religion are spending more on hospitalisation as compared to Muslim households’ Is this difference significant? Explore possible reasons for the difference. Does this finding has any policy implications in Indian context.

6. The authors state that ‘In public facilities, households using unsafe drinking water are reporting a higher burden of OOPE expenditure compared to the households using safe drinking water. This is because the chances of infection are more in the case of former than in the latter.’ NCDs do not occur as a result of infection. Please clarify the statement how it relates in the context of current research.

7. Please include a section on limitations of your study

Conclusion: Please discuss appropriate policy implications of your findings

Reviewer #2: 1. Introduction:

• There are some spelling mistakes.

• Some abbreviations are written without the whole word at first such as GDP in page 3

• On the other hand, some phrases are repeated and abbreviations should be used such as DALY use the abbreviation not the Disability Adjusted Life Years and United Nations Sustainable Development Goals in page 3.

• The estimated figure in page 3.. what figure?

2. Study design:

• An NCD should be a NCD.

• Use the abbreviation for non-communicable disease

3. Results:

• Table 1: I think a column for the total % should be added

• INR: I understood it is for Indian Rupee but the whole word was not written anywhere in the manuscript

• Table 3: I think you can test the significant difference between different variables and put the p value

• Comment for table 3 should be before the table.

• Table 4: how is the percentage calculated? It is not equal 100%

• Table 5: why there is mention for 10% significant? Mostly we use 1% or 5% only. 10% consider not-significant

4. Discussion:

• Why there is a table 6 in the discussion? It should be in the result section and it is not present in the manuscript at all.

5. Conclusion:

• Use abbreviations for non-communicable disease, out-of-pocket expenditure and catastrophic health expenditure

6. Ethical consideration:

• I don't understand why there is no ethical approval or statement?

6. PLOS authors have the option to publish the peer review history of their article (what does this mean?). If published, this will include your full peer review and any attached files.

Reviewer #1: No

Reviewer #2: No

---

## [Author Response · Author response to Decision Letter 0]

27 Oct 2021

Response to reviewers

Title: “Uneven economic burden of non-communicable diseases among Indian households: A comparative analysis"

 Ref: PONE-D-21-18147

At the outset, the authors express their deep sense of gratitude to the Editor-in-Chief of Plos One and anonymous reviewers for their valuable suggestions that helped to improve the literal and technical content of this manuscript. Response to the reviewers comments are marked in blue colour.

Reviewer: 1

Comments to the Corresponding Author

Thanks for providing me the opportunity to review this manuscript. The authors analyze the Indian national survey NSSO 75th round. The analysis has value in terms understanding the economic burden of NCDs in India. This research could provide reflections on how the recent healthcare reforms in lines with India’s UHC aspirations has succeeded/failed to produce coveted effects. Presently the analysis and overall manuscript stand week. There are several lacks related to methodological description, presentation of descriptive results, theoretical basis of choice of variables in regression, focused discussion, policy implications etc. This manuscript can benefit from major revision. The specific comments for each section are given below.

Response: Authors are thankful to the anonymous reviewers for appreciation. The concerns of the reviewer have been addressed in a most effective way as listed out point-wise below:

Abstract

1. The term ‘lower and middle income countries’ is incorrect, it is not clear whether the authors are referring to ‘low and middle income countries’ or lower-middle income countries’ please revise accordingly.

Response: Revised as suggested.

2. Please provide the acronym-NCDs at its first use.

Response: Revised as suggested.

3. Please elaborate the methods used.

Response: Thank you so much for the suggestion. We have elaborated the method now.

4. Results: please mention if you are reporting mean expenditures.

Response: As suggested, we have mentioned mean expenditure in the text.

5. Results: please replace ‘2 times more’ with ‘more than twice’.

Response: As suggested, ‘2 times more’ has been replaced with ‘more than twice’

6. Results: The text ‘Particularly, expenditures on medication and diagnosis are troublesome for households with NCDs.’ seems generic. It would be good if authors can provide proportion of total OOP expenditure incurred on medicines and diagnostics.

Response: Thank you so much for the kind observation. As suggested, we have provided proportion of total OOP expenditure incurred on medicines and diagnostics.

7. For results of generalised linear regression model please provide odds ratio and p-values.

Response: We have provided odds ratio and p-values in table-5. If we will provide the same in abstract it will become lengthier as we have to provide for both NCDs and non-NCDs in private and public facilities.

8. Presently, the conclusion statement is crude. It needs to be specific with expansion.

Response: Revised as per suggestion.

Introduction

1. General comments: There are multiple grammatical errors in the text. I have attempted to point out many of these mistakes, but a thorough copy-edit is required to catch all of these errors. Additionally, the introduction is unnecessarily lengthy owing to repetition of concepts. The structure of the introduction needs to be reworked to ensure that distinct ideas are grouped together into paragraphs and that concepts flow to build a context across paragraphs.

Specific comments:

1. Page 2 line 3, as explained above please replace the term ‘lower and middle income countries’

Response: Revised as suggested.

2. Page 2 line 3 please replace the word ‘kills’ with ‘kill’.

Response: As suggested, we have replaced the word.

3. Page 2 line 4 please pluralize the word ‘death’.

Response: As suggested, we have pluralized the word.

4. Please be consistent in the use of the acronym for non-communicable diseases. There is an inconsistency in the use of the full form and the acronym. Additionally, at some places ‘NCD’ is used and at others it is ‘NCDs’.

Response: Thank you so much for your suggestion. We have revised this now.

5. Page 2 line 9 please pluralize the verb ‘trap’

Response: As suggested, we have pluralized the word.

6. Page 2 line 10 please replace the word like with ‘such as’

Response: As suggested, we have replaced the word.

7. Page 2 line 11 again the authors have used the full form instead of the abbreviation for non-communicable diseases. It is advised to give the full form at first-in-text reference along with the acronym in parenthesis and then use the acronym in the subsequent references.

Response: Revised as suggested.

8. Page 2 line 13 please revise the sentence ‘the economic cost of NCDs.’ for it to be grammatically correct. Additionally, please specify ‘…macroeconomic effect’ for whom?

Response: Revised as suggested.

9. Page 2 line 14 please replace the word ‘force’ with ‘workforce’

Response: Replaced the word as suggested.

10. Page 3 lines 1-2, there is incorrect use of prepositions, please revise accordingly.

Response: Revised as per suggestion.

11. Page 3 lines 4-5, please do not capitalize the words succeeding colon and semi colon.

Response: Revised as suggested.

12. Page 3 line 15 please pluralize the word ‘death’

Response: As suggested, we have pluralized the word.

13. Page 3 line 15 please add ‘the’ before the word ‘...death are’

Response: Added ‘the’ as suggested.

14. Page 3 Please do not provide the full form for DALYs again. Also look out for similar errors for other acronyms (UHC, SDG) used in the manuscript.

Response: Made correction as suggested.

15. Page 3, the authors provide DALYs from 1990-2016, it is advised to provide more recent figures using GBD study 2019.

Response: Thank you for your observation. But we are unable to provide more recent figure because in GBD study 2019, DALY has given for particular NCD, not for NCDs as a whole that we have given in our manuscript.

16. Page 3 please provide a reference for the sentence ‘OOPE constitutes 58.7% of total health expenditure in India’

Response: References have been provided in the text.

17. Page 3 the figures cited by the authors ‘23 to 32 million people below the poverty line’ does not refer to impoverishment due to NCDs. Furthermore, the reference cited uses data from 1999-2000 which is more than two decades old. The authors are advised to accordingly revise these facts.

Response: Revised as per suggestion.

18. Page 3 please pluralize the word ‘policymaker’

Response: As suggested, we have pluralized the word.

19. Page 3 it is not clear how understanding of OOP expenditure would provide information on prevention and control of NCDs. Additionally, how would this reduce financial hardship. These lines need additional clarifications.

Response: We have revised these lines. 

20. Page 4 contrary to the authors’ claim there are multiple studies assessing the OOP expenditure for NCDs apart from the two cited. Please provide a strong justification of the novelty of your study and how would it contribute to existing literature.

Response: Incorporated as suggested.

METHODOLOGY

General comments: The authors have analysed nationally representative NSSO data and provide useful descriptive results. The findings are important but empirically obvious. Furthermore, the depth of analysis is thin, the authors do not justify the methodology for computation of catastrophic health expenditures and the theoretical basis of selection of independent variables for their regression model.

Specific comments:

1. The authors do not provide a clear justification on how have they grouped the 15 broad disease categories into NCDs and non-NCDs. For instance, respiratory ailments can be either NCDs or infectious diseases.

Response: We have grouped the disease into two categories, based on the International classification of diseases (ICD-10), where the first category includes only NCDs and the rest of all infectious/communicable/ injuries/nutritional related diseases are grouped into the second category i.e. non-NCDs. And regarding respiratory ailments, some of them are NCDs for.eg like asthma and some of other respiratory ailments like acute upper respiratory infections (cold, runny nose, sore throat with cough, allergic colds) have been classified as non-NCDs. 

2. Page 5: under variables section it is not clear why the authors choose these particular variables in this analysis. For instance, why do the authors think religion is an important variable impacting your outcome variables-OOP expenditure due to NCDs. Similarly, when you have already included wealth quintile as an independent variable why do you include type of ‘latrine’ which is a component of estimating proxy of economic status of household.

Response: As reported in the past studies, socio-economic variables have a significant impact on pattern of healthcare spending, especially in Indian Context. Therefore, this paper tries to understand the socioeconomic differentials in mean OOPE on NCDs and non-NCDs households in public and private health facilities. So, accordingly, we have considered caste, religion along with all other socio-economic and demographic variables that are available in the data set. Regarding the use of both wealth quintile and type of latrine used, it has clearly mentioned in NSSO report that the calculation of wealth quintile is based on household’s usual monthly consumer expenditure and type of latrine is not a component of it. 

3. Page 6, the authors state that they use world bank criteria to measure OOPE. It is not clear what is meant by this since OOPE is already reported in NSSO data.

Response: The methods of calculating OOPE is initially suggested by world bank and it is a globally adopted method. NSSO has also adopted the same method what we are using. 

4. The authors do not provide a justification of calculating CHE based on 10% of total income. Previous literature reports that using consumption expenditure is a better indicator for calculating CHE in comparison to income. Further a range of thresholds is used 10%, 25%, 30% and 40%, please justify the selection of 10% cut-off.

Response: Thank you for your suggestion. We have mistakenly written income instead of consumption expenditure. Because in NSSO monthly consumption expenditure has been reported (see Q-16, D-3, scheduled 25.0 in NSSO report). Now we have corrected it. And regarding cut off point for CHE, though many previous studies have used 10%, 25%, 30% and 40% cut-off, the present study uses 10% threshold for calculating CHE because it is the most widely used threshold level, particularly, the United Nation SDG 3.8.2 has used 10% as its cut-off level (reference no. 33, 46, 55, 56).

5. Page 6 para 2 there is repetition of what is given in the preceding text.

Response: We have deleted the repetition part.

6. Page 6 para 3, the authors provide theoretical description of a generalized linear model which may not be needed. Instead as mentioned previously please consider adding details on the reason for selection of variables in your model. Further, there is no mention about how multicollinearity issue was assessed and addressed?

Response: We have deleted the theoretical description of a generalized linear model and incorporated other necessary changes as suggested.

Results

1. For results ‘Households hospitalised due to NCDs have more elderly members as compared to the non-NCD’ please mention corresponding figures-what proportion

Response: Incorporated as suggested.

2. ‘Nearly 80% of households, both NCDs and non-NCDs’ this is obvious depending on overall NSSO sample and India’ demographic statistics where majority of the population is Hindu.

Response: Yes, it is obvious. But, we are just trying to give descriptive statistics of our study. 

3. Page 7, lines 4-5 please the authors report results for social class for NCDs households, please report corresponding results for Non-NCDs households as well.

Response: Incorporated as suggested.

4. Page 7 lines 6 the authors report combined results for NCD and non-NCDs households, please report the findings separately.

Response: Now we have reported the findings separately for NCD and non-NCDs household.

5. Also reporting results for all categories is redundant for example the authors state 96% drink safe water so it is not necessary to state that 4% are drinking unsafe water. It is suggested to report major findings only.

Response: Text is revised as per suggestion.

6. Please provide corresponding numbers ‘More number of hospitalisation cases are found in the case of households having 1-5 members as compared to 6-7 and 8+ members’ and also how does it differ in NCDs versus non-NCDs households.

Response: Incorporated as suggested.

7. Page 7 in the sentence ‘As expected, analysis of hospitalisation.’, the authors should refrain from making speculations.

Response: Revised the sentence as per suggestion.

8. Overall, please use a standardized pattern for reporting descriptive results. For some variables the authors provide compare results for both NCDs and non-NCDs households and for others overall results for the complete sample are provided.

Response: Incorporated as suggested.

9. Page 9 lines 1-5, please provide results as proportion of total expenditure rather than absolute numbers.

Response: Incorporated as suggested.

10. The authors report Hindus are spending more than Muslims for NCDs is this difference statistically significant?

Response: Results of OOPE are not statistically significant, we have provided significance level for CHE only.

11. There is an inconsistency in reporting results for OOP expenditure, for some variables the authors make an intravariable comparison (Hindus versus Muslims) for others the authors compare differences for NCDs vs non-NCDs households.

Response: We have revised it now.

12. Table 2 and 3 please reconsider the title of the table ‘Difference in mean out-of-pocket expenditure’, the authors do not report the difference in these tables.

Response: Title of the table 2 and 3 has been revised.

13. The authors report CHE is less for others categories for public hospitals and more for private hospitals with Hindus as reference category. Have the authors controlled for confounders for this analysis. Also in your discussion section please provide plausible reasons why this difference is observed.

Response: Yes, we have controlled for confounders for this analysis and also provided plausible reason for why this difference is observed.

.

14. Why is the role of gender not explored for OOPE and CHE. Previous literature has identified gender as one of the major independent variables affecting OOP and CHE.

Response: Our study is based on household level analysis, so we cannot consider gender as an independent variable because it represents an individual characteristic. So, we have taken into consideration of only household level variables. 

Discussion

General comments: The discussion section largely repeats the results of analysis and the authors do not comprehensively discuss the reasons for observing specific findings. 

Specific-comments

1. Page 18 line 1, the findings differ from what has been reported in results section and Table 1

Response: We have deleted it as it is a repetition in results section in Table 1 and we have revised the sentence as well.

2. Page 18 sentence ‘poor people are not going for seeking care due to cost constraints’, please reference this appropriately.

Response: Appropriate references are incorporated.

3. Page 18 sentence ‘highest number of hospitalisation cases reported from the southern region and lowest from the northeast’. Please discuss the basis of this finding.

Response: Now we have discussed this finding.

4. ‘In public health centres, the rural-urban difference in mean OOPE is found to be very less among NCD households, however, this difference is more while considering non-NCD households. Why? Please discuss.

Response: Now we have discussed this finding.

5. ‘households that belong to the Hindu religion are spending more on hospitalisation as compared to Muslim Households’ Is this difference significant? Explore possible reasons for the difference. Does this finding has any policy implications in Indian context.

Response: Possible reason has been provided for this finding. Yes, it has a policy implication in Indian context. As Muslims have lower education level (as per evidence by previous literature), they may not able to grab the available insurance facilities provided by Indian government. So, government must increase public awareness regarding the availability of the insurance facility. Along with this the government should also take necessary steps to improve the education of the Muslim community.

6. The authors state that ‘In public facilities, households using unsafe drinking water are reporting a higher burden of OOPE expenditure compared to the households using safe drinking water. This is because the chances of infection are more in the case of former than in the latter.’ NCDs do not occur as a result of infection. Please clarify the statement how it relates in the context of current research.

Response: Though the main focus of this study is to calculate the burden of NCDs, we are also comparing the burden of NCDs with non-NCDs households and as per the previous literature drinking water is an important factor affecting communicable/infectious disease. So, we are using drinking water just as a socio-economic variable in our model with the hypothesis that unsafe drinking water will affect more to non-NCDs households than NCDs households.

7. Please include a section on limitations of your study

Response: Limitations have been added in the study.

Conclusion: Please discuss appropriate policy implications of your findings

Response: Incorporated as suggested. 

Reviewer: 2

Comments to the Corresponding Author

1. Introduction:

• There are some spelling mistakes.

Response: Now we have made the correction.

• Some abbreviations are written without the whole word at first such as GDP in page 3

Response: Full form of GDP has been given.

• On the other hand, some phrases are repeated and abbreviations should be used such as DALY use the abbreviation not the Disability Adjusted Life Years and United Nations Sustainable Development Goals in page 3.

Response: Revised as suggested.

• The estimated figure in page 3. what figure?

Response: Revised it now.

2. Study design:

• An NCD should be a NCD.

Response: Revised as suggested.

• Use the abbreviation for non-communicable disease

Response: We have used as suggested.

3. Results:

• Table 1: I think a column for the total % should be added.

Response: Thank you so much for the kind observation. A column for the total % have been added in the text.

• INR: I understood it is for Indian Rupee but the whole word was not written anywhere in the manuscript

Response: Thank you for your suggestion. Now we have written it right below the table.

• Table 3: I think you can test the significant difference between different variables and put the p value

Response: Since you have asked for significant difference between different variables, now we have run ANOVA for that and accordingly given p value. Results in bold letter are significant at 5% in table 2 and 3.

• Comment for table 3 should be before the table.

Response: Revised as suggested.

• Table 4: how is the percentage calculated? It is not equal 100%

We have calculated percentage separately for each group. For e.g. we want to see how many people from rural area are facing CHE due to hospitalization, and the same for urban area also. As per our result (Table 4), 30.24% rural people are exposed to catastrophic health payment and the rest 69.76% are not experiencing the same. Similarly, in urban area 23.85% are experiencing CHE due to hospitalization.

• Table 5: why there is mention for 10% significant? Mostly we use 1% or 5% only. 10% consider not-significant

Response: Thank you so much for the suggestion. Now we have removed that.

4. Discussion:

• Why there is a table 6 in the discussion? It should be in the result section and it is not present in the manuscript at all.

Response: Revised as suggested.

5. Conclusion:

• Use abbreviations for non-communicable disease, out-of-pocket expenditure and catastrophic health expenditure

Response: Revised as suggested.

6. Ethical consideration:

• I don’t understand why there is no ethical approval or statement?

Ethical approval is not applicable because the study has used publicly available unit-level data from a secondary source.

---

## [Decision Letter · Decision Letter 1]

15 Nov 2021

Uneven economic burden of non-communicable diseases among Indian households: A comparative analysis

PONE-D-21-18147R1

Dear Dr. BEHERA,

We’re pleased to inform you that your manuscript has been judged scientifically suitable for publication and will be formally accepted for publication once it meets all outstanding technical requirements.

Kind regards,

Petri Böckerman

Academic Editor

PLOS ONE

Additional Editor Comments (optional):

Reviewers' comments:

Reviewer's Responses to Questions

**Comments to the Author**

1. If the authors have adequately addressed your comments raised in a previous round of review and you feel that this manuscript is now acceptable for publication, you may indicate that here to bypass the “Comments to the Author” section, enter your conflict of interest statement in the “Confidential to Editor” section, and submit your "Accept" recommendation.

Reviewer #2: All comments have been addressed

2. Is the manuscript technically sound, and do the data support the conclusions?

Reviewer #2: Yes

3. Has the statistical analysis been performed appropriately and rigorously? 

Reviewer #2: Yes

4. Have the authors made all data underlying the findings in their manuscript fully available?

Reviewer #2: Yes

5. Is the manuscript presented in an intelligible fashion and written in standard English?

Reviewer #2: Yes

6. Review Comments to the Author

Reviewer #2: (No Response)

7. PLOS authors have the option to publish the peer review history of their article (what does this mean?). If published, this will include your full peer review and any attached files.

Reviewer #2: No

---

## [Editor Report · Acceptance letter]

2 Dec 2021

PONE-D-21-18147R1 

Uneven economic burden of non-communicable diseases among Indian households: A comparative analysis 

Dear Dr. Behera:

I'm pleased to inform you that your manuscript has been deemed suitable for publication in PLOS ONE. Congratulations! Your manuscript is now with our production department. 

Kind regards, 

on behalf of

Professor Petri Böckerman 

Academic Editor

PLOS ONE